# Enhancement of Proton Conductivity Performance in High Temperature Polymer Electrolyte Membrane, Processed the Adding of Pyridobismidazole

**DOI:** 10.3390/polym14071283

**Published:** 2022-03-22

**Authors:** Kehua Lin, Chengxiang Wang, Zhiming Qiu, Yurong Yan

**Affiliations:** School of Materials Science and Engineering, South China University of Technology, 391 Wushan Road, Guangzhou 510641, China; linkh@yyjnd.com (K.L.); bao220212@163.com (C.W.)

**Keywords:** proton exchange membrane, fuel cell, pyridine, imidazole

## Abstract

A pyridobisimidazole unit was introduced into a polymer backbone to obtain an increased doping level, a high number of interacting sites with phosphoric acid and simple processibility. The acid uptake of poly(pyridobisimidazole) (PPI) membrane could reach more than 550% (ADL = 22), resulting in high conductivity (0.23 S·cm^−1^ at 180 °C). Along with 550% acid uptake, the membrane strength still held 10 MPa, meeting the requirement of Proton Exchange Membrane (PEM). In the Fenton Test, the PPI membrane only lost around 7% weight after 156 h, demonstrating excellent oxidative stability. Besides, PPI possessed thermal stability with decomposition temperature at 570 °C and mechanical stability with a glass transition temperature of 330 °C.

## 1. Introduction

Fuel cells showed great promise in energy storage, especially in the onboard automotive industry and stationary power generation [1,2,3,4,5,6,7,8]. However, carbon monoxide was easily absorbed on the catalyst during the reformation of hydrocarbon, which becomes a dominant obstruction for the further use of fuel cells. To avoid the high cost of purified hydrogen, many new routes were developed to produce fuel cells. Among these, the High Temperature Polymer Electrolyte Membrane Fuel Cells (HTPEM FCs) are the most attractive due to the permission of higher operation temperature [9,10,11]. Studies also indicated that the higher operating temperature not only reduced the absorption of carbon monoxide but also simplified the management of fuel cells in water management and the generation of waste heat [4,11,12,13,14,15].

Generally, the higher operating temperature of HTPEM FCs presented a higher demand for the thermal, mechanical and long-term stability of the membrane. It was found that PBI is most widely used in HTPEM for its exceptional stability under serious conditions. To improve the proton conductivity, phosphoric acid (PA) was always intermingled as a proton carrier, and the proton transport deeply depended on the ratio of PBI/phosphoric acid, namely, the acid doping levels (ADLs: defined as the number of moles of doping phosphoric acid per mole PBI repeat unit). Meanwhile, the imidazole rings of PBI were a major binding site of phosphoric acid, which provided the proton transfer channel and prevented acid leaking. However, two major challenges still hindered the wide use of acid-doped PBI in HTPEM FCs, that is, the poor solubility of the polymer and the low proton conductivity. To improve the solubility of this material, researchers attempted to introduce bulky groups and flexible groups into its backbone to regulate the ADL, and asphenylindan [16], naphthalene [17], and fluorine [18,19] were proven to be useful [16,20,21]. As for the conductivity, disclosures indicated that the conductivity of the PBI membranes was not only related to the ADL but also influenced by the number of proton acceptor sites [22,23,24]. Reports that focused on incorporating more proton acceptor sites, such as imidazole [21,25,26], pyridine rings [25,27,28], and nitro [29] into the monomer structure demonstrated a higher proton conductivity. For example, Han et al. grafted Trifluoromethanesulfonylimide on PBI, and found that the higher proton concentration provided by TFSI with stronger acidity contributed much to proton conductivities [6]. Kim et al. studied H_3_PO_4_-doped cross-linked benzoxazine−benzimidazole copolymer membranes, showing 0.12 S/cm proton conductivity at 150 °C. The hydroxyl and tert-amine of polybenzoxazine provided proton acceptance sites and polybenzoxazine inherent stability [30]. Liu et al. prepared polybenzimidazole containing hydroxyl groups (PBI-OH)/ionic-liquid-functional silica (ILS) composite membranes without phase separation. Additionally, the ILS enhanced the ADL and proton conductivity (0.106 S/cm at 170 °C) [31]. Mohamed R. Berber synthesized bipyridine-based polybenzimidazole (Bipy-PBI) polymers, pointed out the N atom of pyridine provided loading of phosphoric acid molecules [32].

In this study, we firstly synthesized pyridobisimidazole-contained polymer for HTPEM fabrication and named it poly(pyridobisimidazole) (PPI). The pyridine ring in the polymer backbone is expected to improve the solubility [33,34]. Groups, such as NH or N on the pyridine that hold low pKa values would be a benefit for the increase of acid doping levels and conductivity of the membrane. The continuous nitrogen atom of the pyridobisimidazole unit maybe remarkably increase the proton acceptor sites that favor phosphoric acid doping and enhance the proton conductivity. To confirm our suspicions, the structure of PPI has been characterized by FT-IR and ^1^H-NMR. Its thermal property was evaluated by thermogravimetric analysis (TG). The morphology of the membrane was depicted by scanning electron microscope (SEM). Oxidative stability, mechanical properties, water uptake, swelling ratio, ADL, inherent viscosity as well as conductivity were also expounded to explore its prospects in the fuel cell.

## 2. Experimental

### 2.1. Materials

4,4′-oxobisbenzoic acid (98%) was purchased from Maya regent (Jiaxing, China) and purified by recrystallization, which was performed in a ratio of ethanol to DMF 3:1 at 140 °C. 2,6-diaminopyridine was obtained from Guangzhou Kaffin biotechnology Co., Ltd. (Guangzhou, China) and used as received. TAP·3HCl·H_2_O was obtained from 2,6-diaminopyridine according to the literature [35]. Polyphosphoric acid (PPA), phosphoric anhydride (P_2_O_5_), *N, N*-dimethyl acetamide (DMAc), *N*,*N*-dimethylformamide (DMF), *N*-methyl-2-pyrrolidone (NMP), dimethyl sulfoxide (DMSO), phosphoric acid (phosphoric acid), methane sulfonic acid (MSA) and tetrahydrofuran (THF) were all commercially obtained from Macklin Reagent Co., Ltd. (Guangzhou, China), and used as received. Deionized water (DI) was made by our laboratory.

### 2.2. Synthesis of PPI

PPI was prepared by condensation polymerization based on TAP·3HCl·H_2_O and 4,4′-oxybisbenzoic acid. The process was shown in Figure 1. TAP·3HCl·H_2_O (6.00 g, 22.5 mmol), PPA (100 g), and tin powder (0.20 g, used as a reducing agent) were charged into a 150 mL three-necked flask, which was equipped with nitrogen-purge in and out and mechanical stirrer. At first, the doughy mixture was stirred at 50 °C to remove hydrogen chloride (HCl) for an hour. Then the temperature was raised to 90 °C to further get rid of the HCl in the reaction system for another hour. After that, 4,4′-oxybisbenzoic acid (5.81 g, 22.5 mmol) and P_2_O_5_ (3.00 g) were added to it. The reaction temperature was maintained at 100 °C for 1 h, 120 °C for 4 h, 140 °C for 12 h, and 180 °C for 10 h. Afterward, the dark red viscous mixture was directly transferred to DI to remove Phosphoric acid. The obtained polymer was further boiled thoroughly with DI until neutral in pH and dried up in a vacuum oven at 100 °C for 12 h. In the end, a dark green powder was obtained and stored under dry conditions for further usage.

### 2.3. Membrane Preparation

The membranes were prepared by a solution casting method. At first, PPI was dissolved in MSA at 100 °C for 12 h, the bubbles in the solution were removed by ultrasonic, and then a homogenous solution was obtained. Afterward, the solution was cast on a clean petri dish and heated at 140 °C for 20 h. The petri dish was cooled down to room temperature and then immersed into deionized water to peel off the membrane automatically. The membrane was boiled with water three times and then dried in a vacuum oven at 100 °C for 12 h. The thicknesses of the membranes were controlled between 80~100 μm.

### 2.4. Measurements

Fourier Transforms Infrared Spectra (FT-IR) of PPI membranes and phosphoric acid-doped PPI membranes were scanned using a Nicolet iS5 which was equipped with an attenuated total reflectance infrared spectroscopy (ATR-FTIR, Thermo Scientific, Waltham, MA, USA). The measurements were carried out at ambient temperature from 4000–650 cm^−1^ with 32 times scans at a resolution of 4 cm^−1^. The inherent viscosity (I.V.) of the polymer was determined on 0.05 g·dL^−1^ concentration of PPI dissolved in MSA with an Ubbelohde capillary viscometer at 25 ± 0.1 °C. ^1^H nuclear magnetic resonance (NMR) spectroscopy data were obtained with an NMR spectrometer Avance III HD 600 (Bruke, Zürich, Switzerland).

The thermal stability of the as-prepared polymer was evaluated by thermogravimetric analysis (TGA) with a Netsch system (Selb, Germany). The membrane was dried at 180 °C in a vacuum oven for 12 h to remove the absorbed water and residual organic solvents before testing. Afterward, the samples were tested from 30 to 800 °C with a heating rate of 20 °C·min^−1^ under both air and N_2_ atmosphere.

The Differential Scanning Calorimeter was performed on a Netsch DSC 204F1 (Selb, Germany). The sample was preheated at 140 °C for 12 h. The measurement was carried out at a heating rate of 10 °C·min^−1^ in N_2_.

The solubility of the polymer was studied by dissolving 5 mg polymer in 0.5 mL solvent. The membrane was immersed into a different solvent for 4 h at 25 °C. Additionally, the insoluble solvents of PPI at 25 °C would be carried out at 100 °C for 2 h.

The PPI membranes were immersed into different concentrations of phosphoric acid solutions. The pristine volume (V_0_) and weight (W_0_) of the membrane were recorded. After the membrane was immersed in the acid for 12 h at room temperatures, they were removed from the acid solution and a filter paper was used to remove the residual acid on its surface. Then the membranes were dried at 110 °C for 12 h and the weight of the membranes (W_doped_) was recorded as soon as possible.

The ADL of a membrane which was defined as the number of phosphoric acids per repeat unit of PPI was calculated with the mass difference of the original (W_0_) and doped (W_doped_) membrane samples. The ADL of a membrane was calculated by the following formula and the small number of dissolved polymers was ignored:ADL = (W_doped_ − W_0_)/M_PA_/(W_0_/M_PPI_)(1)

M_PA_ is the molecular weight of phosphoric acid, M_PPI_ represents the molecule weight of the repeat unit of PPI.

The dimension stability of the membranes was characterized by swelling ratio (SR) and calculated by the following formula:SR = (V_wet_ − V_0_)/V_0_ × 100%(2)

The morphology of undoped and phosphoric acid-doped membranes was obtained by an Evo 08 scanning electron microscope (SEM). The cross-section was obtained by fracturing the samples in the liquid nitrogen. All the samples were sprayed with Au before the measurement.

The oxidative stability was determined by the weight loss of samples after being immersed into a Fenton agent. At first, a pre-weighed dry membrane was submerged in a 50 mL Fenton solution with 3 wt% H_2_O_2_ and 4 ppm Fe^2+^ (added as FeSO_4_·7H_2_O) at 68 °C. Then, for every 12 h, the sample was taken out, washed three times with deionized water, dried at 110 °C for 12 h, and recorded the weight immediately. Fenton agent would be refreshed every time for continued testing.

The mechanical properties of the doped and undoped membranes were performed on a Zwick tensile machine (BT1-FR010TH.A50, Germany) under an ambient atmosphere. All the samples were cut into 5 mm × 30 mm and dried in a vacuum oven at 110 °C for 12 h before the test. Afterward, they were measured at a speed of 2 mm·min^−1^ and each sample was carried out at least four times to obtain a reliable average value.

The proton conductivity of the phosphoric acid-doped PPI membranes was measured by four-probe alternating current (AC) impedance spectroscopy over a frequency range of 1–10^5^ Hz from 20 to 180 °C using an electrochemical workstation (CHI 660 E). All the samples (1 cm × 4 cm) were dried at 110 °C for 12 h before the test. Then they were sandwiched between two pairs of polytetrafluorethylene (PTFE) plates and measured in anhydrous conditions. The proton conductivity (σ) was calculated by the given formula:σ = L/RA u(3)

L refers to the distance between the two inner electrodes, R represents the R_s_ (resistance) value obtained from the membrane and A indicates the cross-sectional area of the membranes.

The activation energy (Ea) denotes the minimum energy necessary for ion transport through membranes [8]. Ea was calculated from Arrhenius plots by the given formula: lnσ = lnσ_0_ − Ea/RT(4)
where σ and σ_0_ are the proton conductivity and pre-exponential factor, respectively, in mS cm^−1^, Ea is the activation energy necessary for protons to transport in kJ mol^−1^, R is the gas constant in kJ mol^−1^, and T is the absolute temperature in K.

## 3. Results and Discussion

### 3.1. Structural Analysis

The structures of PPI and acid-doped PPI were confirmed by FT-IR and ^1^H NMR, as shown in Figure 1. The characteristic peaks of N–H at 3187 cm^−1^ were observed, while no peaks were observed at 1680 cm^−1^, demonstrating the complete cyclization of the imidazole ring. The absorption peaks around 1036 and 1287 cm^−1^ were assigned to asymmetrical C–O–C stretching vibration, which indicated the successful introduction of the ether band in the backbone of the polymer. The peak at 1165 cm^−1^ was attributed to the stretching vibration of C=N. Hence, PPI was successfully synthesized through the above results. The broad peak of 2500–2250 cm^−1^ was assigned to –OH stretching [36]. The broad peak at 1300–800 cm^−1^ indicated the vibration of hydrogen phosphate and phosphoric acid groups. As shown in Figure 1b, the peak at 9.5 ppm was assigned to the aromatic proton of the hydroxy-phenyl ring a, while the peak at 7.8 ppm was assigned to b. The peak at 8.6 ppm was assigned to the proton of the pyridine ring. While the absence of the proton of the imidazole ring in Figure 1b could be ascribed to the interaction between the SO_4_^2−^ and N–H groups [37].

### 3.2. Thermal Stability

The thermal stability of proton exchange membranes plays a vital role in HT-PEM. As shown in Figure 2a, under both air and N_2_ atmosphere, the initial decomposition temperature is above 560 °C; 5% weightless temperature both reach 550 °C. As shown in Figure 2b, The mass loss below 500 °C could be attributed to the removal of adsorbed water(50–150 °C) and phosphoric acid [38] ( started at 180 °C). As displayed in Table 1, the inherent viscosity value of PPI is 6.4 dL·g^−1^ in MSA at 25 ± 0.1 °C, indicating the success of polymerization and demonstrated a high molecular weight polymer have been obtained. The high glass transition temperature (T_g_) at 329.9 °C proves the rigid structure of PPI. The working temperature of HTPEM FCs ranges from 80–200 °C. The high decomposition temperature and T_g_ ensure the usage of PPI in the HTPEM.

### 3.3. Solubility

Many aromatic polymers, such as PBI, PBT, and PBO, have limited applications due to their poor solubility and processing ability. To solve this problem, we introduced ether bonds and pyridine rings to obtain a higher inherent viscosity value (6.4 dL·g^−1^) polymer. Table 2 lists the solubility properties of PPI in solvents. The as-prepared polymer dissolved conveniently in proton acid owing to the introduction of pyridine ring compared to m-PBI (η_inh_./dL g^−1^ = 2.45) and p-PBI (η_inh_./dL g^−1^ = 3.22) [39]. What´s more, it is also soluble in DMSO and NMP by heating without any additives. The solubility of PPI in H_3_PO_4_ makes it easier to prepare high ADL membranes for PEM.

### 3.4. Acid Doping and Swelling Ratio 

Figure 3a showed the ADL of PPI membranes after immersion in phosphorous acid solution at 25 °C for 12 h. The acid uptake was experienced in two stages. Firstly, the ADL increases slowly while the phosphoric acid concentrations increase from 10 wt% to 60 wt%. According to the previous literature [40,41,42], the nitrogen atoms with two unpaired electrons on the imidazole ring and the pyridine ring can easily interact with phosphoric acid [35,43]. Secondly, ADL increases abruptly between 70 wt% and 75 wt% concentrations of phosphoric acid, thereby ADL reached 6 and 22, respectively. Previously reported by J. Lobato [36] excess phosphoric acid could be easily introduced into the polymer chains at high concentrations of phosphoric acid, leading to a high volume between the polymer chains for phosphoric acid uptake. These results demonstrate the N atoms on the pyridine rings can improve the acid affinity.

The small swelling ratio of the membrane due to the existing strong interaction between the polymer chains was demonstrated in Figure 3b. While under the high phosphoric acid concentration, the phosphoric acid molecules between polymer chains could weaken the interaction between polymer chains, leading to the volume inside of the membrane increasing dramatically.

### 3.5. Surface Morphology

The surface and cross-section morphologies of the membranes were investigated by SEM. Figure 4a–d showed the surface morphology of PPI membranes become much more uneven with ADL of the membranes increasing. When the concentration of phosphoric acid solution reached 60%, some PPI dissolved in phosphoric acid solution due to the incorporation of pyridine. On one hand, high ADL membranes for PEM are ascribed to the introduction of pyridobisimidazole. However, the strong hydrogen bonding between –NH– and –N= groups would be broken during the preparation of acid-doping membranes, leading to an uneven surface, the swelling ratio increasing and the mechanical property of the membrane decreasing.

### 3.6. Oxidative Stability

During the long-term operation of PEM, hydrogen peroxide or hydroxyl radicals damage the nitrogen-containing heterocyclics on the polymer backbone. To study the oxidative stability of membranes, the Fenton test, a simulation condition under the practical working conditions of PEMs was carried out in this study. The as-prepared samples were immersed into Fenton reagent (3% H_2_O_2_ solution containing 4 ppm of Fe^2+^, refreshed every 12 h) at 68 °C. At the same time, the weight loss and the breaking time of the samples were recorded, as presented in Figure 5.

The weight decreased gradually as time went on, and the pure PPI membranes displayed a weight loss of around 7% after 156 h of the Fenton test. While m-PBI membranes progressively broke into small pieces in 60 h [44], PPI membranes remained integrated after 156 h in the Fenton test. The nitrogen atom on the pyridine unit reacts with radicals by forming pyridine-N-oxide, which will hamper the oxidation reactions of imidazole groups, leading to the high durability of the membranes [43].

### 3.7. Mechanical Properties

The mechanical property of the membrane plays an important role in the performance of PEM. Figure 6 shows the mechanical strength of the doped and un-doped PPI membranes at ambient conditions. The tensile strength of the un-doped PPI membrane reached up to 137 MPa equivalent to that of the PBI membrane. It could be attributed to the strong hydrogen bonding interactions between the polymer chains, and the rigid structure of the polymer. As for the doped membranes, the tensile strength decreased with the acid-doping level increasing, while elongation corresponds to the trend of ADL. It could be due to the plasticization of the phosphoric acid molecules that exist between the polymer chains. The tensile of the membrane with a doping level of 22 still reached 10 MPa. It could be attributed to the higher hydrogen bond density caused by the nitrogen atoms on the pyridine rings interacting with –NH– groups [45]. Compared to the phosphoric acid-doped PBI membrane, which showed 6.7 MPa at a doping level of 14.5, acid-doped PPI membranes showed much better mechanical properties [44].

### 3.8. Proton Conductivity

The proton conductivity of the membrane was carried out in the temperature range of 20~180 °C under anhydrous conditions, as shown in Figure 7a. The proton conductivity of the acid-doping PPI membranes was positively related to temperature. High temperature increases the reaction activity, thereby reducing the charge-transfer resistance of fuel cells [46]. When the doping level reached 22, the membrane showed the highest proton conductivity and even up to 0.23 S·cm^−1^ (180 °C). The proton conductivity of the membranes had a linear relationship with the acid-doping level due to the number of H^+^ and the free volume of the membrane increased with acid-doping level. The PBI with ADL 3.1 showed a proton conductivity around 0.001 S·cm^−1^, while the PPI membranes with ADL 3 showed a proton conductivity around 0.01 S·cm^−1^. It has been studied that the conjugate acid of imidazole possessed a pK_a_ value of 7.18 [10], while the pyridine conjugate acid was 5.25 [47]. Qian et al. introduced fluorine into PBI with 30-40ADL (phosphoric acid) only showed conductivity of 0.04 S/cm at 190 °C. In Figure 7b, the Ea of all membranes ranged from 7.10–14.15 kJ/mol, demonstrating excellent mobility of H^+^ ions of the acid-doping membranes. In all, the incorporation of pyridine rings could not only increase the ratio of phosphoric acid molecules existing between the polymer chains but also enhanced the proton conductivity of the membranes by increasing the number of proton acceptor sites.

### 3.9. Performance Indicator (PI)

Both proton conductivity and mechanical properties are important for the performance of PEM, but there is a trade-off effect between them. In this study, a performance indicator (PI) is proposed to describe the overall performance of proton conductivity and mechanical properties:PI = PA conductivity (mS·cm^−1^) × tensile strength (MPa) × 10^−3^(5)

Roughly, PI values <0.1, 0.1~0.5, and >0.5 indicate poor, intermediate and potentially good materials, respectively. As shown in Table 3. Three kinds of acid-doping PPI showed PI values higher than 0.5. The PPI-3 showed the highest PI values (1.227) indicating PPI itself has great potential.

## 4. Conclusions

A novel polymer, PPI was synthesized successfully. With the incorporation of the pyridine unit, it possessed good solubility in solvents. PPI membranes were made by the solution casting method. Additionally, a high ADL membrane was obtained due to the basic N atoms on the pyridine rings. The thermal stability of PPI was confirmed higher than 550 °C. The Fenton test showed that the PPI membrane possessed excellent oxidative stability, which was attributed to the introduction of pyridine units. PPI-22 membrane with high proton conductivity of 0.23 S·cm^−1^ at 180 °C was obtained. PPI is a very promising material for the next generation of HTPEM.

## Data Availability

The data presented in this study are available on request from the corresponding author.

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
