# Peer review of "Enhancement of Proton Conductivity Performance in High Temperature Polymer Electrolyte Membrane, Processed the Adding of Pyridobismidazole"

_polymers, 2022, doi:10.3390/polym14071283_

Round 1
Reviewer 1 Report
In this paper, the authors reported the results of fabricating and characterizing poly(pyridobisimidazole) (PPI) membranes for high-temperature proton exchange membrane fuel cell applications. Although this paper reports interesting results, in order to be published in "polymers", many parts need to be revised from the point of view of the completeness of the paper. My detailed comments are as follows:
- The title must be revised. There are many sentences with grammatical problems, and overall correction is required by a native speaker.
- After defining initially, do not mix the terms "ADL" and "acid doping level", but use only "ADL".
- From the results of Figure 3, it can be seen that ADL is increased at 60 wt% or more of PA concentration, but water uptake and swelling ratio are also greatly increased. At the actual operating temperature (e.g. >180 oC), it can be expected that these figures become significantly larger. Ultimately, there is a question about whether the fabricated PPI membranes have the proper physical stability that can be used under actual operating conditions. From the result of Figure 6, it can be seen that the tensile strength is significantly decreased with the increase of ADL, and it is necessary to prove whether the PPI membranes have sufficient strength even at high temperatures. In this regard, please provide the differential scanning calorimetry (DSC) results as well.
- The fuel cell application results (MEA test) must be included in the paper.
- The journal names indicated in the references should be abbreviated.
Author Response
Dear Reviewer 1:
Thank you for the comments concerning our manuscript entitled “Pyridobisimidazole Improves the Properties and Application of High Temperature Proton Exchange Membrane”. Those comments are all valuable and very helpful for revising and improving our paper, as well as the important guiding significance to our researchers. We have studied comments carefully and have made correction which we hope to meet with approval. Please see the attachment.

Reviewer 2 Report
The manuscript titled “Pyridobisimidazole improves the properties and application of high-temperature proton exchange membrane” by Yurong Yan et al. This article addressed the pyridobisimidazole synthesis, membrane preparation, structural, morphological, and electrochemical characterization. The authors provided a well-prepared presentation, and the present work is more interesting and needs major revision before publication in the “Polymers”
- In the introduction section, the authors should discuss the recent achievements of pyridobismidazole polymer-based high-temperature PEM fuel cell applications.
- The authors can recheck Figure 3a. Why did the author measure the water uptake in the present studies? Because present studies reveal the high-temperature PEM fuel cell application hence only a need for phosphoric acid doping.
- The authors may include IEC results and should compare the acid doping and swelling results.
- In figure 7, the authors should calculate the activation energy for prepared PPI using the Arrhenius equation.
- In section 3.6, the authors should provide the chemical stability for different doping level membranes.
- The authors should provide the differential scanning calorimetry (DSC) results for prepared membranes.
- Some of the important recent references need to cite in the suitable place of the revised introduction or analysis section for better providing information, such as DOI: 1016/j.energy.2021.121791; 10.1016/j.compositesb.2021.108828; 10.1021/acsami.9b18059; 10.1016/j.jpowsour.2009.11.021
- The typographical errors should be corrected in all sections of the revised manuscript.
- The authors should follow the Polymers journal reference style.
Author Response
Dear Reviewer 2:
Thank you for the comments concerning our manuscript entitled “Pyridobisimidazole Improves the Properties and Application of High Temperature Proton Exchange Membrane”. Those comments are all valuable and very helpful for revising and improving our paper, as well as the important guiding significance to our researchers. We have studied comments carefully and have made correction which we hope to meet with approval. Revised portions are marked in red in the paper. Please see the attachment.

Round 2
Reviewer 1 Report
The authors have carefully revised the manuscript according to the referees’ comments. In my opinion, this manuscript could be accepted for publication in polymers. However, the English should be further improved and the title should be revised more appropriately.
Author Response
Dear Reviewer 1:
Thank you for the comments concerning our manuscript Those comments are all valuable and very helpful for revising and improving our paper, as well as the important guiding significance to our researchers. We have studied comments carefully and have made correction which we hope to meet with approval. Revised portions are marked in red in the paper. The main corrections in the paper and the responds to the reviewers’ comments are as flowing:
Response to Reviewer1 Comments
Point 1: The authors have carefully revised the manuscript according to the referees’ comments. In my opinion, this manuscript could be accepted for publication in polymers. However, the English should be further improved and the title should be revised more appropriately.
Response 1: We are very sorry for our poor English language proficiency and incorrect writing. We have made correction according to the Reviewer’s comments. We have re-check the whole paper and marked the correction in red in the paper. And we have renamed this title as “Enhancement of Proton Conductivity Performance in High Temperature Polymer Electrolyte Membrane, Processed the Adding of Pyridobismidazole.” This title was written with reference to Christopher Koenigsmann’s ”Size-dependent enhancement of electrocatalytic performance in relatively defect-free, processed ultrathin platinum nanowires”. DOI: 10.1021/nl100718k
Reviewer 2 Report
This manuscript is advised to be accept since all the problems have been addressed.
Author Response
Dear Reviewer 2:
Thank you for your recognition.